# Magnetic Resonance Neuroimaging in Amyotrophic Lateral Sclerosis: A Comprehensive Umbrella Review of 18 Studies

**DOI:** 10.3390/brainsci15070715

**Published:** 2025-07-03

**Authors:** Sadegh Ghaderi, Sana Mohammadi, Farzad Fatehi

**Affiliations:** 1Neuromuscular Research Center, Department of Neurology, Shariati Hospital, Tehran University of Medical Sciences, Tehran 1416634793, Iran; mohammadi.sana@iums.ac.ir (S.M.); f-fatehi@sina.tums.ac.ir (F.F.); 2Department of Neuroscience and Addiction Studies, School of Advanced Technologies in Medicine, Tehran University of Medical Sciences, Tehran 1416634793, Iran; 3Neurology Department, University Hospitals of Leicester NHS Trust, Leicester LE5 4P, UK

**Keywords:** amyotrophic lateral sclerosis, functional MRI, diffusion MRI, perfusion MRI, susceptibility-weighted imaging, quantitative susceptibility mapping

## Abstract

**Background/Objectives:** Despite extensive research, the underlying causes of amyotrophic lateral sclerosis (ALS) remain unclear. This umbrella review aims to synthesize a vast body of evidence from advanced magnetic resonance imaging (MRI) studies of ALS, encompassing a wide range of neuroimaging techniques and patient cohorts. **Methods:** Following the PRISMA guidelines, we conducted an extensive search of four databases (PubMed, Scopus, Web of Science, and Embase) for articles published until 3 December 2024. Data extraction and quality assessment were independently performed using the AMSTAR2 tool. **Results:** This review included 18 studies that incorporated data from over 29,000 ALS patients. Structural MRI consistently showed gray matter atrophy in the motor and extra-motor regions, with significant white matter (WM) atrophy in the corticospinal tract and corpus callosum. Magnetic resonance spectroscopy revealed metabolic disruptions, including reduced N-acetylaspartate and elevated choline levels. Functional MRI studies have demonstrated altered brain activation patterns and functional connectivity, reflecting compensatory mechanisms and neurodegeneration. fMRI also demonstrated disrupted motor network connectivity and alterations in the default mode network. Diffusion MRI highlighted microstructural changes, particularly reduced fractional anisotropy in the WM tracts. Susceptibility-weighted imaging and quantitative susceptibility mapping revealed iron accumulation in the motor cortex and non-motor regions. Perfusion MRI indicated hypoperfusion in regions associated with cognitive impairment. **Conclusions:** Multiparametric MRI consistently highlights widespread structural, functional, and metabolic changes in ALS, reflecting neurodegeneration and compensatory mechanisms.

## 1. Introduction

Amyotrophic lateral sclerosis (ALS) is a fatal neurodegenerative disease that causes a range of debilitating symptoms including muscle weakness, atrophy, and eventual paralysis and death [1]. ALS can affect both the motor and non-motor regions of the brain [2] and spinal cord [3]. Although the underlying causes of ALS remain elusive, researchers have made significant strides in understanding its complex nature using magnetic resonance imaging (MRI).

MRI techniques provide a non-invasive, biomarker-based, and quantitative means to examine the brain, enabling detailed visualization and quantification of structural, functional, metabolic, perfusion, and both micro- and macro-structural changes associated with ALS [4]. These techniques have uncovered a range of abnormalities in patients with ALS, including cortical atrophy, degeneration of gray matter (GM) and white matter (WM), and alterations in functional connectivity (FC) within specific brain regions. Beyond conventional structural MRI (sMRI), advanced techniques such as sMRI-based analysis [5,6], diffusion tensor imaging (DTI) [7], functional MRI (fMRI) [8], magnetic resonance spectroscopy (MRS) [9], susceptibility-weighted imaging (SWI) [10,11], perfusion-weighted MRI [12], and quantitative susceptibility mapping (QSM) [13,14] are valuable tools for detecting subtle changes in the brain using neuroimaging approaches.

Previous systematic reviews and meta-analyses have explored MRI biomarkers in ALS; however, these works often focus on isolated modalities, limiting their capacity to capture the disease’s complexity. Moreover, existing reviews frequently prioritize motor-related changes, underrepresenting the role of extra-motor regions and their clinical correlations [15,16].

Thus, this review aims to provide a comprehensive, multi-modal synthesis of MRI findings in ALS, critically evaluating the strengths and limitations of current evidence while proposing standardized imaging protocols to enhance diagnostic accuracy, prognostic stratification, and therapeutic monitoring in clinical practice.

## 2. Materials and Methods

We employed an umbrella review, a comprehensive evidence synthesis method, to summarize the existing systematic neuroimaging reviews in the context of ALS. This review adhered to the Preferred Reporting Items for Systematic Reviews and Meta-analyses (PRISMA) 2020 guidelines [17]. The methodology drew upon both the Cochrane protocol for review overviews and the Joanna Briggs Institute approach for umbrella reviews [18,19]. Additionally, we followed the “Overviews of Reviews” methodology outlined by Cochrane Training accessed on 3 December 2024 (https://training.cochrane.org/handbook/current/chapter-v). Figure 1 represents a summary of the Section 2, including the search strategy, eligibility criteria, study selection, data extraction, assessment of methodological quality, and the possibility of meta-analysis.

### 2.1. Search Strategy

We searched four databases (PubMed, Scopus, Web of Science, and Embase) for articles published up to 3 December 2024, without language restrictions. We employed Boolean operators (AND and OR) and relevant MeSH terms and keywords to construct our search strategy. Detailed search strategies for each database are provided in Appendix A.

### 2.2. Eligibility Criteria and Study Selection

This review focuses on systematic reviews and meta-analyses examining MRI biomarkers in patients with ALS to synthesize evidence on the utility of neuroimaging in understanding disease mechanisms. The included studies were systematic reviews or meta-analyses of primary research evaluating MRI-derived measures, such as brain atrophy, WM integrity, and FC, in adults diagnosed with ALS according to established criteria, comparing these measures to those in healthy controls. Reviews with full-text publications in peer-reviewed journals were considered without restrictions on publication date. Exclusion criteria included non-systematic reviews (e.g., narrative and scoping reviews), original articles, commentaries, editorials, quasi-experimental studies, studies focusing on non-MRI neuroimaging techniques, pediatric populations, and non-peer-reviewed sources. The eligibility criteria were the Population, Exposure, Comparator, and Outcome (PECO) framework [20], focusing on patients with ALS (population), MRI as the primary exposure, healthy controls as the comparator, and neuroimaging findings as the outcome. Records were independently screened and selected by two reviewers using EndNote software (v21), and any disagreements were resolved through discussion and consensus with the other author (professor of neurology). The screening process included both title and abstract reviews in the initial phase and full-text evaluations in the subsequent phase, adhering to PRISMA 2020 guidelines to ensure methodological transparency and rigor.

### 2.3. Data Extraction

Reviewers (SG and SM) independently extracted essential data from the included systematic reviews and meta-analyses, ensuring comprehensive and accurate organization of results. Subsequently, the primary reviewer (SG) verified and coordinated the accuracy of the extracted data to maintain consistency and reliability. The data are summarized in Table 1, which provides detailed information on the following aspects: study identification (study and year), country of the first author, search timeframe, databases searched, number of included studies, types of included studies, and number of patients and controls. It also specifies whether a meta-analysis was performed, the quality assessment methods employed (including publication bias evaluations), and any additional analyses conducted (e.g., statistical techniques). Furthermore, Table 1 highlights the brain regions of interest (ROIs) investigated, region-specific MRI findings, and main findings related to MRI biomarkers, including the techniques and metrics utilized.

### 2.4. Assessment of Methodological Quality

Two reviewers (SG and SM) independently assessed the methodological quality of the selected systematic reviews and meta-analyses was assessed independently by two reviewers (SG and SM) utilizing the Assessment of Multiple Systematic Reviews 2 (AMSTAR2) tool [21]. The AMSTAR2 tool requires evaluators to provide responses classified as “No”, “No Meta-analysis”, “Partial Yes”, or “Yes”. Based on the evaluation, the overall quality of the assessments was categorized into four levels: “High quality”, “Moderate quality”, “Low quality”, or “Critically low quality”. This instrument comprises 16 items distributed across seven critical domains (Appendix A).

### 2.5. Meta-Analysis

In this umbrella review, no new statistical analyses were conducted, as the primary aim was to critically evaluate the strength of the existing evidence and identify research gaps within the field. The review included 18 studies, including 13 systematic reviews and 6 meta-analyses, all of which provided valuable insights into MRI biomarkers in ALS. These studies were carefully selected to ensure a comprehensive coverage of the literature, focusing on diverse aspects of ALS pathophysiology, including structural, functional, and connectivity-based MRI findings.

## 3. Results

### 3.1. Study Selection

This umbrella review included both meta-analyses and systematic reviews. Figure 2 illustrates the selection process. An initial literature search, constrained by publication date, identified 16,781 papers: 3023 records in PubMed, 4405 in Scopus, 2041 in the Web of Science, and 6952 in Embase (Appendix A). After removing 7750 duplicates, the remaining papers underwent title and abstract screening, resulting in the exclusion of 8846 records. Subsequently, 185 papers were selected for a comprehensive full-text review based on their relevance to the research topic. Ultimately, 18 studies were included in this umbrella review, as detailed in Table 1 and Appendix A, and Figure 2. Moreover, a comprehensive summary of all findings derived from the Section 3, as illustrated in Figure 3.

### 3.2. Summary of Findings

The umbrella review included 18 studies that utilized databases such as PubMed, Scopus, Web of Science, Ovid EMBASE, MEDLINE, CINAHL, and Cochrane to search for relevant articles. In total, these studies incorporated data from over 29,000 patients with ALS. Five of these studies were meta-analyses [27,34,36,38,39]. All but seven [28,29,31,35,36,37,38] of the studies reported the guidelines used for conducting and reporting the systematic review. However, the reporting of guidelines and methods for quality and bias assessment were inconsistent across the studies. The quality assessment methods employed included the Newcastle–Ottawa Scale (NOS) [23,25,27], a modified version of the National Heart, Lung, and Blood Institute quality assessment tool for case-control studies [32], Egger’s regression test and rank correlation test [27], Cochrane risk of bias tool [24,25], and Quality Assessment of Diagnostic Accuracy Studies (QUADAS) [38].

### 3.3. Structural MRI

SMRI studies in individuals with ALS have consistently demonstrated GM atrophy in various brain regions. The precentral gyrus, a crucial motor area, exhibited substantial volume loss [29,30,33,35]. This atrophy often extends to other frontal lobe regions [35], such as the orbitofrontal [31], opercular [31], and cingulate cortex, particularly anterior part [24,29,31,33,39]. The temporal cortex [35], particularly the hippocampus [22,24,33,35], parahippocampal gyrus [22,24], and amygdala [24,33], which are involved in memory and emotional processing, also shows atrophy [22,24,33]. Furthermore, GM atrophy is observed in the insula [29,30,35], parietal cortex [35], and less frequently, the occipital lobe and cerebellum [34,35]. Thalamic atrophy is a common finding, especially in ALS patients with frontotemporal dementia (ALS-FTD), primary lateral sclerosis (PLS), or C9orf72 mutation [23,31]. WM atrophy is another prominent finding in sMRI studies of ALS [33]. Significant alterations were observed in the primary motor pathway, which showed significant alterations [31]. The corpus callosum (CC), which connects the two hemispheres of the brain, also exhibits WM atrophy [31,33]. Additional WM tracts affected include the frontotemporal connections, middle cerebellar peduncles, and uncinate fasciculus [24].

One promising aspect of fractal dimension (FD) analysis is its ability to detect subtle structural changes that may not be readily apparent using conventional MRI techniques. FD in the motor cortex was positively correlated with ALSFRS-R scores, a widely used measure of functional impairment in ALS [29]. Furthermore, FD in the motor cortex was negatively correlated with the rate of disease progression, indicating that patients with more rapid disease progression exhibited more pronounced reductions in cortical complexity. Another study reinforced these findings by demonstrating that FD values correlate with clinical metrics such as ALSFRS-R scores, disease progression rates, and the presence of frontotemporal dementia [30]. Their study further strengthens the notion that FD analysis could serve as a valuable tool for assessing disease severity, monitoring disease progression, and predicting cognitive decline in patients with ALS.

Texture analysis involves extracting quantitative features from medical images that describe the spatial arrangement and heterogeneity of the pixel intensities. By applying texture analysis to MRI images of the CST and CC, researchers have been able to distinguish ALS patients from HCs, suggesting that texture features may capture subtle microstructural alterations associated with the disease [33]. This emerging technique has the potential to develop sensitive and specific imaging biomarkers for the diagnosis and prognosis of ALS. Furthermore, they highlighted the importance of a semi-quantitative analysis of studies to identify the most commonly reported regions with abnormalities in each MRI modality. This analysis can provide a comprehensive list of candidate regions to be included as features in machine learning models for disease classification in ALS. This suggests that a multimodal approach combining various MRI techniques and clinical data could significantly enhance the performance of machine-learning models for ALS diagnosis, prognosis, and personalized treatment strategies.

### 3.4. Magnetic Resonance Spectroscopy

MRS studies in ALS have demonstrated significant metabolic alterations in various brain regions, primarily indicating neuronal loss or dysfunction, and glial activation. The most consistent finding is a reduction in N-acetylaspartate (NAA), a marker of neuronal integrity, in motor areas such as the motor cortex and corticospinal tract (CST) [26,33,35]. This reduction in NAA, often reported as a decreased NAA/creatine (Cr) ratio, has been observed in both cross-sectional and longitudinal studies, suggesting ongoing neuronal loss throughout the disease course [33,35].

Elevated levels of choline (Cho) are frequently observed in regions where NAA is reduced [35]. This increase in Cho is thought to reflect glial activation or inflammation, processes that accompany neuronal damage in ALS. MRS studies have also revealed metabolic alterations in extra-motor regions, particularly in the hippocampus. Patients with ALS show significantly elevated levels of total NAA (tNAA), tNAA/tCr, and total Cho (tCho) in the hippocampus compared to healthy controls, indicating a complex interplay of neuronal loss, glial activation, and metabolic dysfunction in this region [22,26]. Furthermore, changes in other metabolites, such as glutamate (Glu) [22,26] and myo-inositol (mIn) [33], have been reported in the hippocampus and precentral gyrus of ALS patients. These alterations suggest further metabolic disturbances and potential involvement of glial cells in the pathophysiology of ALS. Additionally, sodium–MRS studies have shown higher total sodium concentrations in the right precentral gyrus and CST in patients with ALS [33].

### 3.5. Functional MRI

fMRI studies in patients with ALS have revealed significant alterations in brain function, particularly in the motor and extra-motor networks. Task-based fMRI studies have shown both increased and decreased activation of various brain regions during motor tasks [35,36]. Some studies have reported increased activation in the motor cortex and other regions, suggesting compensatory mechanisms to maintain motor performance despite neuronal loss [35,36]. Other studies have shown decreased activation, which may reflect progressive loss of motor neurons and impaired motor function [24].

Resting-state fMRI, which measures the FC between brain regions at rest, has also revealed significant alterations with ALS. Studies have consistently reported reduced FC in motor networks [23,24,28,33,35], including the primary motor cortex and the premotor cortex [28]. This decreased connectivity within motor networks likely reflects the disruption of neural communication and coordination caused by the degeneration of motor neurons with other regions, such as the thalamus [23].

Furthermore, altered FC is observed in extra-motor networks [22,23,28], including the default mode network (DMN) [28] and the frontoparietal network [28]. The DMN, which is involved in self-referential and introspective processes, often shows increased connectivity in patients with ALS, which may reflect compensatory mechanisms or alterations in cognitive function. Longitudinal studies have shown that FC changes progress over time, with a progressive increase in FC in the frontoparietal and frontostriatal networks, correlating with disease severity [28]. This suggests that brain networks undergo dynamic reorganization in response to disease progression, potentially involving compensatory mechanisms and maladaptive changes.

Specific memory-related tasks assessed using fMRI have revealed altered brain activation patterns in patients with ALS [22,24]. Working memory tasks show increased activation in frontotemporal and parietal regions [24]. Novelty processing tasks exhibit increased activation in the hippocampus [22,24]. Memory storage tasks reveal decreased activation in the prefrontal cortex, occipital cortex, and fusiform gyrus, alongside increased activation in the superior frontal areas, suggesting a disruption in the top-down regulation of memory storage [24].

ALS involves extra-motor cortex areas, causing cognitive dysfunction and deficits in the socioemotional and sensory processing pathways [36]. Tasks involving emotional decision-making and memory recognition show enhanced activation in the left hemisphere and reduced activation in the right hemisphere, implicating the posterior cingulate gyrus in the interplay between emotional and memory processes [24]. Language-related tasks, such as letter fluency and confrontation naming, demonstrate decreased activation in the prefrontal, temporal, parietal, and occipital areas, particularly in the inferior frontal gyrus [24]. These findings highlight the impact of ALS on cognitive function beyond motor control [36,39].

Cerebellar function is also affected in ALS [24,28], as evidenced by fMRI studies. Cerebellar resting-state signal fluctuations show an inverse correlation with changes in digit span forward performance, suggesting a role for the cerebellum in cognitive function [24]. Furthermore, functional abnormalities in the Papez circuit, which is a neural pathway involved in memory function, are associated with memory performance [24].

### 3.6. Diffusion MRI

Diffusion MRI, particularly DTI, has emerged as a powerful tool for investigating microstructural changes in the brain, providing valuable insights into ALS pathology. DTI studies in ALS consistently demonstrate alterations in WM integrity, primarily characterized by reduced fractional anisotropy (FA), a measure of the directionality of water diffusion in WM tracts [23,28,33,35,39].

The CST, a crucial motor pathway connecting the brain to the spinal cord, shows the most prominent FA reduction in ALS patients [27,33,39]. This finding is consistent across multiple studies and reflects the degeneration of motor neurons and the breakdown of WM structures as CST hyperintensities [27,33] and microstructural alterations [23,28,33,35,39] along the CST. FA reduction in the CST often extends to other motor-related WM tracts, including the corticorubral/corticopontine and corticostriatal tract [33].

DTI studies have also revealed FA reductions in WM tracts beyond the motor system [23,28,33,35,39], indicating that ALS affects a broader network of brain regions. The CC, the major WM tract connecting the two brain hemispheres, consistently exhibits reduced FA, particularly in the motor and premotor segments [24,31,33,35]. Other affected tracts include the frontotemporal connections, middle cerebellar peduncles, uncinate fasciculus, and proximal portion of the perforant path [24,40]. These findings highlight the multisystem nature of ALS and the involvement of both motor and non-motor pathways [39].

In addition to reduced FA, DTI studies in ALS also report increased mean diffusivity (MD) and radial diffusivity (RD) in various WM tracts [22,23,35]. Increased MD reflects a general increase in water diffusion, which can be caused by a variety of factors including axonal loss, demyelination, and inflammation. Specifically, increased RD suggests damage to the myelin sheath surrounding axons, leading to increased water diffusion perpendicular to axon direction [22,23].

DTI metrics, particularly FA, correlate with clinical features and disease progression in ALS [28,35]. Lower FA values are associated with greater disease severity, faster progression rates, and poorer clinical outcomes [35], similar to FD results [29,30]. These reductions were observed in the motor cortex, precentral gyrus, central sulcus, circular sulcus of the insula, cingulate gyrus and sulcus, and post-central gyrus. Moreover, DTI measures in specific brain regions, such as the thalamus, have been shown to correlate with reduced survival, disease severity, disease duration, disease progression, and impaired cognition [23,24,35].

### 3.7. Magnetic Susceptibility and Iron Detection

SWI and QSM are specialized MRI techniques that are particularly sensitive to changes in magnetic susceptibility (χ) within tissues [25,32,33,35]. These susceptibility changes can be caused by various factors including iron accumulation, calcification, and blood products [25,32]. One of the key findings regarding SWI in patients with ALS is the motor band sign (MBS), a hypointense area, sometimes described as a “black ribbon sign”, in the precentral gyrus, the region of the brain responsible for voluntary motor control [25,27]. MBS is attributed to increased iron deposition in deeper layers of the motor cortex. This finding is corroborated by other MRI techniques, such as T2-weighted (T2-W) and T2*-w and QSM [25,32]. The severity of MBS on SWI is correlated with the severity of upper motor neuron (UMN) impairment in ALS patients, highlighting the clinical relevance of this imaging finding [25]. SWI is considered the most sensitive MRI technique for detecting MBS, outperforming other commonly used methods, such as T2-w and FLAIR imaging [25,27].

While the MBS has been extensively discussed in the context of SWI and iron accumulation, it is worth noting that another study also mention the “zebra sign”, [33] another imaging feature observed in the motor cortex of ALS patients. The zebra sign, characterized by a distinct low-signal intensity layer in the precentral cortex, was detected using phase difference enhanced imaging (PADRE) and was observed in approximately 50% of patients with ALS. The specific pathological correlates of the zebra sign and its relationship with MBS warrant further investigation.

QSM is another promising technique for iron detection in ALS patients. The QSM provides a quantitative measure of iron content in brain tissues, going beyond the qualitative assessment offered by the SWI. This quantitative information may provide more sensitive and specific biomarkers for disease diagnosis, monitoring, and therapeutic interventions [25,32]. While MBS is a prominent finding in the motor cortex, iron accumulation in ALS extends beyond this region. QSM studies have consistently shown higher susceptibility in subcortical structures, including the putamen, globus pallidus, red nucleus, and substantia nigra, suggesting increased iron content in these areas [32]. These findings align with the known pathological features of ALS, including TDP-43 pathology and changes in nigrostriatal pathways. Furthermore, the precise role of iron accumulation in ALS pathogenesis remains an active research area. However, several studies have suggested that iron dysregulation may contribute to disease progression of the disease [25,27]. Increased iron levels in neural cells have been linked to SOD1 mutation, a known genetic risk factor for ALS. Moreover, increased oxidative stress, potentially linked to iron, has been implicated in the development and progression of ALS [32].

Ultra-high-field 7T MRI has shown greater sensitivity for detecting MBS in ALS patients than 3T MRI [25,27]. This improved sensitivity is attributed to the higher signal-to-noise ratio and better spatial resolution of 7T MRI, allowing for more detailed visualization of iron-related changes in the brain.

### 3.8. Perfusion MRI

One report showed hypoperfusion in the left parahippocampal gyrus of patients with motor-onset ALS-FTD [22]. The parahippocampal gyrus plays a crucial role in memory encoding and retrieval. Reduced CBF in this area could reflect neuronal loss, synaptic dysfunction, or impaired vascular supply, potentially contributing to the cognitive impairments frequently observed in ALS-FTD. The same study highlighted the potential of CBF measurements in the right hippocampus to differentiate ALS patients from healthy controls. While the specific direction of change (hypo- or hyperperfusion) was not mentioned, this finding suggests that ASL could serve as a valuable diagnostic tool for ALS, particularly in distinguishing the disease from other neurological conditions. Further supporting the involvement of perfusion abnormalities in non-motor areas of ALS, another study observed decreased CBF in the thalamus of ALS-FTD patients [23]. The thalamus acts as a critical relay center for sensory and motor information. Reduced CBF in this region could disrupt these crucial pathways, contributing to both motor and cognitive dysfunctions in patients with ALS-FTD.

### 3.9. Results of Methodological Quality Assessment

Following the guidelines of AMSTAR 2 (Appendix A) and as summarized in Table 2, our assessment of the methodological quality of the included systematic reviews and meta-analyses revealed that nine papers were classified as moderate quality. Additionally, six papers were rated as low quality and three were categorized as critically low quality. These findings underscore the variability in methodological rigor among the studies analyzed, highlighting areas for potential improvement in future systematic reviews and meta-analyses.

## 4. Discussion

This umbrella review, encompassing data from over 29,000 patients with ALS, provides a comprehensive synthesis of MRI findings in ALS. ALS is not confined to motor neuron degeneration, but involves widespread neural dysfunction across the motor and extramotor regions. MRI findings suggest that ALS is a multisystem disorder affecting both the motor and non-motor areas of the brain [41,42,43]. This review highlights the utility of various MRI techniques for characterizing structural, functional, and macro- and microstructural changes associated with ALS. A consistent finding across multiple studies is evidence of widespread GM pathology in ALS, particularly affecting the motor cortex, precentral gyrus, and CST [44,45,46]. These observations align with the established clinical phenotype of ALS, characterized by the progressive loss of motor neurons and subsequent atrophy [16,47]. Beyond the motor system, MRI studies have revealed alterations in the frontal, temporal, and subcortical regions, suggesting a broader involvement of brain networks in ALS [2,48]. This is consistent with the growing recognition of cognitive and behavioral impairments in a significant subset of ALS patients [49]. Advanced MRI techniques, such as voxel-based morphometry and DTI, have proven particularly valuable for demonstrating microstructural and WM degeneration in ALS. Studies have consistently reported reduced FA in the CST, CC, and other WM tracts, indicating the disruption of neuronal connections and impaired signal transmission. fMRI studies add another layer, demonstrating disrupted connectivity within the motor network and between the motor and extramotor areas. In particular, Rs-fMRI has highlighted network alterations in the sensiorimotor, DMN, and language, suggesting that ALS pathology extends beyond motor circuits to include networks associated with cognitive and behavioral functions [50,51,52]. MRS studies have further illuminated the metabolic alterations associated with neuronal loss in ALS. Notably, decreased levels of NAA, a marker of neuronal integrity, have been observed in various brain regions such as the motor cortex and thalamus. These findings highlight the interplay between neurodegeneration and neuroinflammation [53]. T2-w hyperintensities in motor and subcortical WM regions observed in some studies may reflect glial activation and inflammatory changes.

Integrating multiparametric and multimodal neuroimaging approaches can provide a holistic view of the complex interplay between the different pathological processes in ALS. Combining data from various MRI techniques such as sMRI, DTI, MRS, and QSM can offer a comprehensive assessment of structural, microstructural, functional, and metabolic alterations [54,55,56]. The neuroimaging findings reviewed here provide compelling evidence of the involvement of multiple brain regions and networks in ALS. The consistent observation of CST degeneration corroborates the traditional view that ALS is a motor neuron disease. However, evidence of prefrontal atrophy, DMN disruption, metabolic changes, and microstructural alterations in non-motor regions broadens this perspective, suggesting a more widespread neurodegenerative process. These findings are in line with neuropathological studies showing TDP-43 pathology in extramotor regions, supporting the notion that ALS exists on a continuum with other TDP-43 proteinopathies such as FTD [57,58,59,60].

Moving beyond conventional MRI techniques such as DTI for WM alterations, advanced diffusion MRI techniques such as non-Gaussianity-based techniques, diffusion kurtosis imaging (DKI), and neurite orientation dispersion and density imaging (NODDI) provide a more refined understanding of microstructural changes in the WM and GM in ALS. Furthermore, machine learning, texture analysis, and deep learning approaches are emerging as powerful tools for analyzing complex neuroimaging data in ALS. The integration of these approaches with MRI data holds significant potential for improving the diagnostic accuracy, predicting disease progression, and identifying novel imaging biomarkers [61,62]. The heterogeneity of ALS phenotypes, including differences in the site of onset, rate of progression, and genetic background, poses a significant challenge for neuroimaging studies [15,63,64]. Stratification of patients based on clinical and genetic subtypes could enhance the specificity of neuroimaging findings and reveal distinct brain involvement patterns.

## 5. Interpretations for Clinicians

Advanced MRI abnormalities correlate closely with both motor and non-motor clinical manifestations. GM atrophy in the precentral gyrus and CST abnormalities on sMRI align with UMN signs such as spasticity, hyperreflexia, and pathological reflexes. These findings complement clinical assessments, particularly in distinguishing ALS from mimic disorders (e.g., cervical myelopathy) [65], where UMN pathology may be ambiguous. Similarly, DTI reveals microstructural CST degradation (reduced FA) that correlates with faster disease progression, offering prognostic value. Clinicians should note that CST hyperintensities on T2-w imaging or FA reductions on DTI are surrogate markers of UMN dysfunction, independent of the lower motor neuron (LMN)-driven muscle atrophy, thereby aiding phenotypic stratification.

MRS provides metabolic insights critical for early diagnosis. Reduced NAA in the motor cortex and CST reflects neuronal loss and precedes overt atrophy, serving as a sensitive biomarker of disease activity. Elevated Cho in these regions, indicative of glial activation, often coexists with NAA reduction, suggesting concurrent neurodegeneration and neuroinflammation. While longitudinal data on metabolite timing remain limited, cross-sectional evidence implies that NAA decline may occur early, with Cho elevation emerging as gliosis progresses. For clinicians, MRS in motor regions (e.g., the precentral gyrus) could enhance diagnostic confidence, particularly in atypical cases lacking clear UMN signs. Additionally, hippocampal metabolic disturbances (e.g., elevated glutamate) correlate with cognitive decline, underscoring the need for cognitive screening in ALS patients, especially those with C9orf72 mutations or frontotemporal dementia overlap.

To translate these findings into clinical practice, clinicians should prioritize accessible MRI protocols that capture key ALS biomarkers. MRI with T2-w and FLAIR sequences can detect CST hyperintensities, MBS on SWI, and QSM value alterations, all of which correlate with UMN burden. DTI, increasingly available in clinical settings, should be incorporated to assess CST integrity and the condition of other non-motor tracts (as evidenced by reduced FA) to predict progression. For centers with advanced capabilities, MRS of the motor cortex, hippocampus, and dorsolateral prefrontal cortex (DLPFC) can stratify patients by metabolic dysfunction, while resting-state fMRI may identify network disruptions (e.g., within the default mode network, DMN) linked to cognitive decline. Crucially, these tools should complement—not replace—clinical evaluation, as neuroimaging biomarkers enhance diagnostic specificity and prognostic accuracy.

Taken together, MRI is not limited to identifying motor deficits; rather, it provides evidence of both motor and non-motor involvement that may correlate with clinical signs—such as UMN lesions and LMN pathology—leading to progressive muscle weakness. Thus, ALS is a multisystem disorder with structural, functional, and metabolic alterations across motor and extramotor networks. Quantitative imaging biomarkers, such as FD, FA, metabolites, and QSM values, offer objective measures that correlate with clinical severity, progression rates, and cognitive decline, providing a framework for prognostication and therapeutic decision-making. The integration of multimodal MRI data with clinical and genetic profiles holds promise for personalized care. Machine learning models combining sMRI, DTI, and fMRI features may enhance diagnostic accuracy and prognostication, particularly in distinguishing ALS subtypes (e.g., C9orf72 mutation carriers) or predicting rates of progression. Clinicians must also recognize the prognostic implications of extramotor involvement, such as thalamic atrophy or DMN dysfunction, which may necessitate comprehensive neuropsychological assessments and multidisciplinary interventions.

## 6. Conclusions

This umbrella review provides, for the first time, a comprehensive synthesis of MRI findings in the context of ALS, emphasizing the substantial advancements in employing MRI to unravel this complex disease. Consistent GM atrophy in motor regions (such as the precentral gyrus and CST) and extramotor areas (such as the hippocampus and thalamus), WM degeneration in the corpus callosum and frontotemporal tracts, metabolic disturbances (reduced NAA, elevated Cho), and disrupted functional connectivity in motor and default mode networks are observed. Advanced techniques such as FD analysis, DTI, DKI, NOODI, and QSM further reveal microstructural complexity loss, iron accumulation, and dynamic network reorganization, correlating with clinical metrics such as ALSFRS-R scores and disease progression rates. These findings underscore ALS as a multisystem disorder, extending beyond motor pathways to involve cognitive and emotional networks. Integrating multimodal MRI biomarkers—such as DTI for corticospinal tract integrity, SWI/QSM for iron detection, and MRS for metabolic profiling—offers actionable tools to enhance diagnostic specificity, monitor disease progression, and predict cognitive decline.

## Figures and Tables

**Figure 1 brainsci-15-00715-f001:**
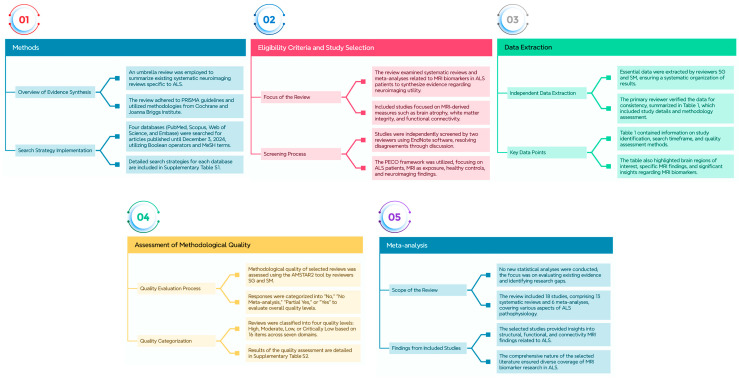
Provides an overview of the Section 2, encompassing the search strategy, eligibility criteria, study selection process, data extraction procedures, evaluation of methodological quality, and the feasibility of conducting a meta-analysis.

**Figure 2 brainsci-15-00715-f002:**
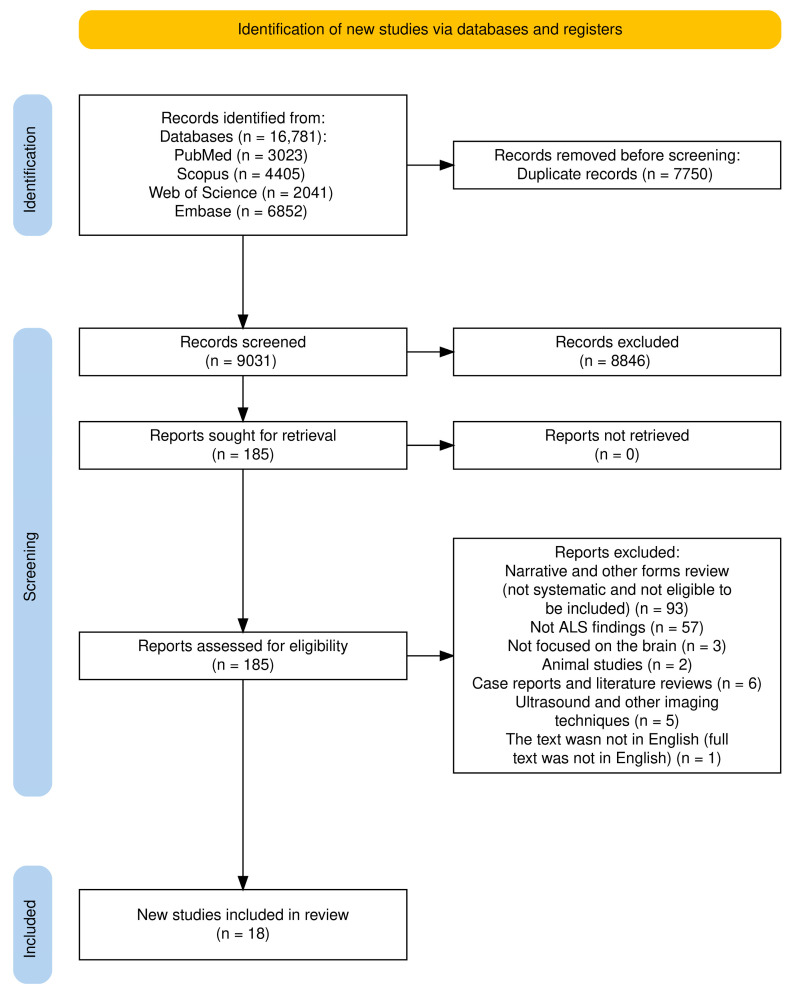
PRISMA flow diagram for systematic review for included studies.

**Figure 3 brainsci-15-00715-f003:**
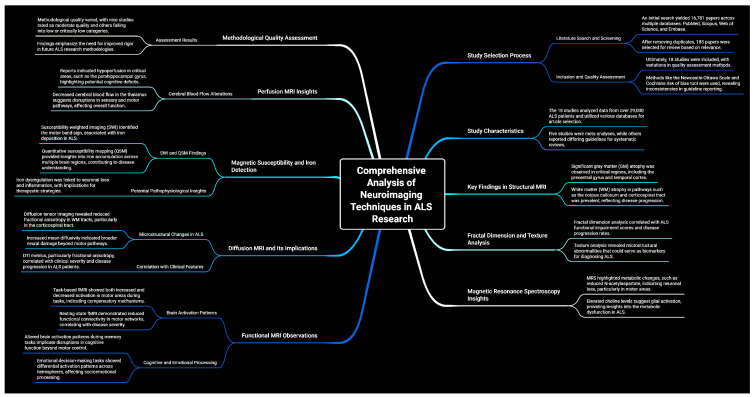
A detailed summary encompassing all key findings.

**Table 1 brainsci-15-00715-t001:** Main characteristics details for included studies (ALS, Amyotrophic Lateral Sclerosis; MOOSE, Meta-Analysis of Observational Studies in Epidemiology; NOS, Newcastle–Ottawa Scale; NR, Not Reported; PRISMA, Preferred Reporting Items for Systematic Reviews and Meta-Analyses; QUADAS, Quality Assessment of Diagnostic Accuracy Studies; ROB, Risk of Bias).

Study, Year	Search Date (Time Frame or Final Search Date)	Databases	Number of Included ALS Studies	Number of ALS Patients *	Meta-Analysis	Guideline	Quality Assessment Method
Mohammadi et al., 2024a [22]	January 2000–July 2023	PubMed and Scopus	46	2156	No	PRISMA 2020	The Cochrane Handbook’s predefined quality assessment criteria were used.
Mohammadi et al., 2024b [23]	Up to June 2023	PubMed and Scopus	52	2386	No	PRISMA 2020	NOS
Ghaderi et al., 2023 [24]	NR	PubMed and Scopus	25	1076	No	PRISMA 2020	The Cochrane Handbook’s predefined quality assessment criteria were used.
Mohammadi and Ghaderi, 2023 [25]	Up to February 2023	PubMed and Scopus	12	285	No	PRISMA 2020	Quality assessment was performed using the NOS for non-randomized studies and the Cochrane risk of bias tool for randomized trials.
Christidi et al., 2022 [26]	Up to January 2022	PubMed	87	1920	No	PRISMA 2020	NR
Zejlon et al., 2022 [27]	Up to 8 April 2021	PubMed, Web of Science and Ovid EMBASE	223; for qualitative synthesis 21; for quantitative synthesis	9152 (specified ALS)	Yes	PRISMA 2015	Egger’s regression test and the rank correlation test were used to assess publication bias. The NOS was used to assess the ROB.
Renga et al., 2022 [28]	NR	PubMed	60	2771	No	NR	NR
Ziukelis et al., 2022 [29]	1980–28 December 2020	PubMed	2	139	No	NR	NR
Meregalli et al., 2022 [30]	Up to 23 March 2021	PubMed and Scopus	2	97	No	PRISMA 2020	NR
Li-Hishing et al., 2021 [31]	Up to May 2020	PubMed	74	>1629	No	NR	NR
Ravanfar et al., 2021 [32]	Up to April 2020	MEDLINE (PubMed interface), Embase (Ovid interface), Scopus, and PsycInfo (Ovid interface).	8	244	No	PRISMA 2009 and 2015	A modified version of the National Heart, Lung, and Blood Institute quality assessment tool for case-control studies was used to assess the risk of bias in each study.
Kocar et al., 2021 [33]	March 2021. The authors excluded publications listed before 1 January 2017, and reviewed articles published in the four years prior.	PubMed	151	1978	No	PRISMA 2020	NR
Gellersen et al., 2017 [34]	Up to 14 July 2016	PubMed	54	1609	Yes	PRISMA	PRISMA
Grolez et al., 2016 [35]	NR	PubMed	116	2305	No	NR	NR
Shen et al., 2015 [36]	Up to April 2015	Ovid Medline, PubMed, and Emabase	55	1086	Yes	NR	NR
Schuster et al., 2015 [37]	Up to June 2014	PubMed	18	412	No	NR	NR
Foerster et al., 2013 [38]	1966–April 2011	MEDLINE, Embase, CINAHL, and Cochrane databases	11	221	Yes	NR	QUADAS
Li et al., 2012 [39]	1990–25 December 2010	PubMed, Web of Science, Embase, and MEDLINE	8	143	Yes	MOOSE guidelines	NR

* The total number of ALS patients is taken into account irrespective of the form or stage of the disease. It is important to note that this figure may vary from the one presented in the original publication.

**Table 2 brainsci-15-00715-t002:** Methodological quality assessment of the selected systematic review and meta-analyses based on AMSTAR 2 checklist (N, No; NM, No meta-analysis; PY, Probably yes; Y, Yes).

Study, Year	Q1	Q2	Q3	Q4	Q5	Q6	Q7	Q8	Q9	Q10	Q11	Q12	Q13	Q14	Q15	Q16	Overall
Mohammadi et al., 2024a [22]	Y	PY	Y	PY	Y	Y	N	Y	Y	N	NM	NM	Y	Y	NM	Y	Moderate
Mohammadi et al., 2024b [23]	Y	PY	Y	PY	Y	Y	N	Y	Y	N	NM	NM	Y	Y	NM	Y	Moderate
Ghaderi et al., 2023 [24]	Y	PY	Y	PY	Y	Y	N	Y	Y	N	NM	NM	Y	Y	NM	Y	Moderate
Mohammadi and Ghaderi, 2023 [25]	Y	PY	Y	PY	Y	Y	N	Y	Y	N	NM	NM	Y	Y	NM	Y	Moderate
Christidi et al., 2022 [26]	Y	N	Y	N	Y	Y	N	Y	N	Y	NM	NM	N	Y	NM	Y	Low
Zejlon et al., 2022 [27]	Y	PY	Y	Y	Y	Y	N	Y	Y	Y	Y	N	Y	Y	Y	Y	Moderate
Renga et al., 2022 [28]	Y	N	Y	N	Y	Y	N	Y	N	N	NM	NM	N	Y	NM	Y	Low
Ziukelis et al., 2022 [29]	Y	N	Y	N	Y	Y	N	Y	N	Y	NM	NM	N	Y	NM	Y	Critically Low
Meregalli et al., 2022 [30]	Y	N	Y	PY	Y	Y	N	Y	N	Y	NM	NM	N	Y	NM	Y	Low
Li-Hishing et al., 2021 [31]	Y	N	Y	N	Y	Y	N	Y	N	Y	NM	NM	N	Y	NM	Y	Low
Ravanfar et al., 2021 [32]	Y	PY	Y	Y	Y	Y	N	Y	Y	Y	NM	NM	Y	Y	NM	Y	Moderate
Kocar et al., 2021 [33]	Y	N	Y	N	Y	Y	N	Y	N	Y	NM	NM	N	Y	NM	Y	Critically Low
Gellersen et al., 2017 [34]	Y	N	Y	N	Y	Y	N	Y	N	Y	Y	N	N	Y	N	Y	Low
Grolez et al., 2016 [35]	Y	N	Y	N	Y	Y	N	Y	N	Y	NM	NM	N	Y	NM	Y	Low
Shen et al., 2015 [36]	Y	N	Y	PY	Y	Y	N	Y	N	N	Y	N	N	Y	N	Y	Moderate
Schuster et al., 2015 [37]	Y	N	Y	N	Y	Y	N	Y	N	Y	NM	NM	N	Y	NM	Y	Critically Low
Foerster et al., 2013 [38]	Y	PY	Y	Y	Y	Y	N	Y	Y	Y	Y	Y	Y	Y	N	Y	Moderate
Li et al., 2012 [39]	Y	N	Y	Y	Y	Y	N	Y	N	N	Y	N	N	Y	N	Y	Moderate

## Data Availability

This article contains all the data produced or analyzed during this investigation. Further inquiries should be forwarded to the corresponding author.

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
