# Peer review of "Magnetic Resonance Neuroimaging in Amyotrophic Lateral Sclerosis: A Comprehensive Umbrella Review of 18 Studies"

_brainsci, 2025, doi:10.3390/brainsci15070715_

Round 1

Reviewer 1 Report

Comments and Suggestions for Authors

The whole paper is a result of umbrella review - from 17 800 reduced to 18 papers. There are always some MRI methods, that are described with reference to ALS. I have some comments:

1.There are many ways of brain imaging - with suggested possibility of motor and non-motor signs of ALS. But there is not specification of pathological MRI findings and clinical signs, symptoms. For clinician - all the review is very theoretical without some racional link to clinic.

2.An example - MRS - in what brain structures is the NAA really high? What is the time sequence of high cholin and high NAA?

3. In conclusion - changes in WM and GM are in correlation with upper motor neuron lesion and progressive muscle weakness, atrophy. But atrophy is not typical for corticospinal tract lesion, it is a peripheral motor neuron lesion. And some recommendation for clinician is missing - nowadays - pyramidal tract, corpus callosum. But some simple accessible MR investigation would be a good step for clinician after such big analysis.

Author Response

We are pleased to submit the revised version of our umbrella review titled “Magnetic Resonance Neuroimaging in Amyotrophic Lateral Sclerosis: A Comprehensive Umbrella Review of 18 Studies” for consideration by “Brain Sciences.” 

We sincerely thank the editor, the editorial office, and the reviewers for their valuable feedback. We believe that the manuscript has been significantly improved and is now well-suited for publication in Brain Sciences. Kindly find attached the reviewer comments along with our point-by-point responses.

Response to Reviewer 1 Comments
The whole paper is a result of umbrella review - from 17 800 reduced to 18 papers. There are always some MRI methods, that are described with reference to ALS. I have some comments:
General point: Thank you sincerely for dedicating your time to reviewing this manuscript. Below, you will find detailed responses, with the corresponding revisions and corrections clearly highlighted or tracked in the resubmitted files.
Comment 1. There are many ways of brain imaging - with suggested possibility of motor and non-motor signs of ALS. But there is not specification of pathological MRI findings and clinical signs, symptoms. For clinician - all the review is very theoretical without some racional link to clinic.
Response 1: We appreciate the reviewer’s insightful comment. To address this, we have expanded the interpretation of MRI findings in the context of clinical signs and symptoms. Kindly see "5. Interpretations for Clinicians."
Comment 2. An example - MRS - in what brain structures is the NAA really high? What is the time sequence of high cholin and high NAA?
Response 2: We thank the reviewer for this question. NAA is typically highest in regions with dense neuronal populations, such as the motor cortex, corticospinal tract, hippocampus, DLPFC, and so on. In ALS, NAA levels are reduced in these areas, reflecting neuronal loss or dysfunction.
Regarding the time sequence of metabolite changes: NAA reduction is an early marker of neuronal integrity loss and is observed in both cross-sectional and longitudinal studies, suggesting it occurs early in the disease course. Elevated Cho, indicative of glial activation or inflammation, often coexists with NAA reduction but may become more prominent as the disease progresses, reflecting ongoing neuroinflammation. Anyway, while longitudinal data is limited, current evidence suggests that NAA decline precedes or coincides with Cho elevation, providing a metabolic signature of neurodegeneration and glial activation.
Nonetheless, we also acknowledge this further in Section 5 and throughout the text.
Comment 3. In conclusion - changes in WM and GM are in correlation with upper motor neuron lesion and progressive muscle weakness, atrophy. But atrophy is not typical for corticospinal tract lesion, it is a peripheral motor neuron lesion. And some recommendation for clinician is missing - nowadays - pyramidal tract, corpus callosum. But some simple accessible MR investigation would be a good step for clinician after such big analysis.
Response 3: We thank the reviewer for highlighting this important distinction. While muscle atrophy is indeed a hallmark of LMN involvement, the MRI findings discussed (e.g., CST hyperintensities, reduced FA on DTI) reflect UMN pathology, which manifests clinically as spasticity, hyperreflexia, and pathological reflexes. To address the reviewer’s request for practical recommendations, we have added Section 5. We hope these responses and revisions address the reviewer’s concerns and provide a clearer link between the neuroimaging findings and their clinical utility.

Reviewer 2 Report

Comments and Suggestions for Authors

Systematic Review: Beyond 29,000 Patients with Amyotrophic Lateral Sclerosis: An Umbrella Review of Magnetic Resonance Neuroimaging Studies

First of all, thank you very much for giving me the opportunity to review this important scientific manuscript.

Despite the many works carried out in this area, this topic remains very relevant. As MRI research improves every year, new opportunities are emerging to look inside the brains of ALS patients. Unfortunately, many pathogenetic mechanisms remain unclear when diagnosing and determining the prognosis of this disease, especially when it comes to damage to upper motor neurons. In this review, the authors studied the latest research in this area and demonstrated the cutting-edge advances of MRI in diagnosing ALS.

The name is very attractive and attracts a lot of attention. however, this title does not correspond to the content of the work. You have researched 16,781 articles and are currently viewing the last 18 articles. If you want to indicate a number, please indicate the number of manuscripts reviewed, not the total number of patients included in these studies.

Introduction:

  • The introduction is very brief and contains information that could be found in any neuroscience textbook. Here you need to indicate previous similar reviews and what is missing in these works to highlight the importance, novelty and necessity of your research.
  • In line 54: “The application of MRI in ALS research extends beyond diagnosis and characterization”. MRI is a diagnostic research method. how can it extend beyond diagnosis and characterization. Please clarify.

The purpose of your research needs to change:

  • ” To the best of our knowledge”. This phrase is unclear and not suitable to describe the purpose.
  • “this review aimed to provide a comprehensive overview of MRI findings in ALS, highlighting the strengths and limitations of the current research and identifying key areas for future investigation”. I like this purpose in this format, short and to the point. But it is necessary to remove the previous phrases: “To the best of our knowledge, this primary umbrella review aims to synthesize a vast body of evidence from MRI studies on ALS, encompassing a wide range of MR neuroimaging techniques and patient cohorts”, otherwise there will be repetition.

I really liked the Materials and Methods section. everything is stated in detail, very scientifically and at the same time in simple, scientific, understandable language. In addition, Figure 1 makes the text easier to understand. however, it is necessary to improve the clarity of this figure. Difficulties arise when reading the text. I think this is not a problem and is easy to fix.

Results:

  • Study selection and summary findings: The sections are described step by step using informative Figures 2 and 3. Table 1 provides background information on the 18 systematic reviews covering this topic. However, the clarity of Figure 3 is very poor.
  • Structural MRI: The authors identified major changes in brain structure that are characteristic of ALS and explain the wide range of central clinical manifestations in ALS.  The most common changes extracted from 18 reviews were GM atrophy in various brain regions: the precentral gyrus, a crucial motor area, exhibited substantial volume loss, frontal lobe regions such as the orbitofrontal, cingulate cortex, temporal cortex , hippocampus and parahippocampal gyrus, amygdala, thalamic structures and cerebellum. WM atrophy of  corpus callosum, the frontotemporal connections, middle cerebellar peduncles, and uncinate fasciculus. Fractal dimension (FD) and texture analysis are other structural changes that are described in detail.
  • Magnetic Resonance Spectroscopy: The section is written very scientifically, incorporating the latest scientific advances. However, it would be interesting and more informative if you indicated the increase or decrease in the specified biochemical structures as a percentage.
  • Functional MRI: MRI studies have revealed significant alterations in brain function, particularly in the motor and extra-motor networks, mode network (DMN) and the frontoparietal network the default.
  • Magnetic susceptibility and Iron detection: One of the key findings on SWI in patients with ALS is the motor band sign (MBS), a hypointense area, sometimes described as a "black ribbon sign," in the precentral gyrus, the region of the brain responsible for voluntary motor control MBS is attributed to increased iron deposition in deeper layers of the motor cortex.
  • Perfusion MRI: hypoperfusion in the left parahippocampal gyrus. Further supporting the involvement of perfusion abnormalities in non-motor areas of ALS, another study observed decreased CBF in the thalamus of ALS-FTD patients.
  • Results of Methodological quality assessment: Table 2 was compiled according to AMSTAR 2 recommendations. Table 3 is clear and simplifies the task for authors and readers.

Overall, the results are written using the latest data, which is of great value to all specialists in this field. However, there are a large number of abbreviations that make it difficult to understand the text and the reader has to look in the text for a complete description of these abbreviations.

The discussion is short and to the point, strictly based on the results obtained, and runs parallel to the results.

In discussion, in lines 378-380 “These findings align with the well-established clinical presentation of ALS, which is characterized by progressive muscle weakness and atrophy”. Muscle atrophy develops when the lower motor neuron is damaged, and the above changes are signs of upper motor neuron damage.

The “Conclusion” logically follows from the results of the study and is fully consistent with the purpose of the study. However, try to shorten it please.

Author Response

We are pleased to submit the revised version of our umbrella review titled “Magnetic Resonance Neuroimaging in Amyotrophic Lateral Sclerosis: A Comprehensive Umbrella Review of 18 Studies” for consideration by “Brain Sciences.” 

We sincerely thank the editor, the editorial office, and the reviewers for their valuable feedback. We believe that the manuscript has been significantly improved and is now well-suited for publication in Brain Sciences. Kindly find attached the reviewer comments along with our point-by-point responses.

Response to Reviewer 2 Comments
Systematic Review: Beyond 29,000 Patients with Amyotrophic Lateral Sclerosis: An Umbrella Review of Magnetic Resonance Neuroimaging Studies
First of all, thank you very much for giving me the opportunity to review this important scientific manuscript.
Despite the many works carried out in this area, this topic remains very relevant. As MRI research improves every year, new opportunities are emerging to look inside the brains of ALS patients. Unfortunately, many pathogenetic mechanisms remain unclear when diagnosing and determining the prognosis of this disease, especially when it comes to damage to upper motor neurons. In this review, the authors studied the latest research in this area and demonstrated the cutting-edge advances of MRI in diagnosing ALS.
General point: We sincerely appreciate your time and effort in reviewing this manuscript and offering valuable feedback. Detailed responses are provided below, with all corresponding revisions and corrections clearly highlighted or tracked in the resubmitted documents.
Comment 1: The name is very attractive and attracts a lot of attention. however, this title does not correspond to the content of the work. You have researched 16,781 articles and are currently viewing the last 18 articles. If you want to indicate a number, please indicate the number of manuscripts reviewed, not the total number of patients included in these studies.
Response 1: We thank the reviewer for this suggestion. The title has been revised to better reflect the content of the work. The new title is:
“Magnetic Resonance Neuroimaging in Amyotrophic Lateral Sclerosis: An Umbrella Review of 18 Systematic Reviews and Meta-Analyses.”
Introduction:
Comment 2: The introduction is very brief and contains information that could be found in any neuroscience textbook. Here you need to indicate previous similar reviews and what is missing in these works to highlight the importance, novelty and necessity of your research.
Response 2: We have expanded the introduction to include a discussion of previous reviews and their limitations.
Comment 3: In line 54: “The application of MRI in ALS research extends beyond diagnosis and characterization”. MRI is a diagnostic research method. how can it extend beyond diagnosis and characterization. Please clarify.
Response 3: We have removed this sentence and revised it completely for clarity.
Comment 4: The purpose of your research needs to change:
” To the best of our knowledge”. This phrase is unclear and not suitable to describe the purpose.
“this review aimed to provide a comprehensive overview of MRI findings in ALS, highlighting the strengths and limitations of the current research and identifying key areas for future investigation”. I like this purpose in this format, short and to the point. But it is necessary to remove the previous phrases: “To the best of our knowledge, this primary umbrella review aims to synthesize a vast body of evidence from MRI studies on ALS, encompassing a wide range of MR neuroimaging techniques and patient cohorts”, otherwise there will be repetition.
Response 4: We have revised the purpose statement as suggested, removing the redundant phrases. The revised purpose now reads:
“Thus, this review aims to provide a comprehensive, multi-modal synthesis of MRI findings in ALS, critically evaluating the strengths and limitations of current evidence while proposing standardized imaging protocols to enhance diagnostic accuracy, prog-nostic stratification, and therapeutic monitoring in clinical practice.”
Comment  5: I really liked the Materials and Methods section. everything is stated in detail, very scientifically and at the same time in simple, scientific, understandable language. In addition, Figure 1 makes the text easier to understand. however, it is necessary to improve the clarity of this figure. Difficulties arise when reading the text. I think this is not a problem and is easy to fix.
Response 5: Thank you for your kind word. We have revised Figure 1 to improve clarity. Kindy check it. 
Results:
Comment  6: Study selection and summary findings: The sections are described step by step using informative Figures 2 and 3. Table 1 provides background information on the 18 systematic reviews covering this topic. However, the clarity of Figure 3 is very poor.
Response 6: We have redesigned Figure 3 to enhance clarity, using a more organized layout, improved color contrast, and clearer labeling.
Comment 7: Structural MRI: The authors identified major changes in brain structure that are characteristic of ALS and explain the wide range of central clinical manifestations in ALS.  The most common changes extracted from 18 reviews were GM atrophy in various brain regions: the precentral gyrus, a crucial motor area, exhibited substantial volume loss, frontal lobe regions such as the orbitofrontal, cingulate cortex, temporal cortex , hippocampus and parahippocampal gyrus, amygdala, thalamic structures and cerebellum. WM atrophy of  corpus callosum, the frontotemporal connections, middle cerebellar peduncles, and uncinate fasciculus. Fractal dimension (FD) and texture analysis are other structural changes that are described in detail.
Response 7: We thank the reviewer for this positive feedback.
Comment  8: Magnetic Resonance Spectroscopy: The section is written very scientifically, incorporating the latest scientific advances. However, it would be interesting and more informative if you indicated the increase or decrease in the specified biochemical structures as a percentage.
Response 8: We sincerely appreciate your kind words. We acknowledge your point; however, due to the heterogeneity in study methods, additional quantitative data was not applicable.
Comment  9: Functional MRI: MRI studies have revealed significant alterations in brain function, particularly in the motor and extra-motor networks, mode network (DMN) and the frontoparietal network the default.
Response 9: We thank the reviewer for this positive feedback.
Comment  10: Magnetic susceptibility and Iron detection: One of the key findings on SWI in patients with ALS is the motor band sign (MBS), a hypointense area, sometimes described as a "black ribbon sign," in the precentral gyrus, the region of the brain responsible for voluntary motor control MBS is attributed to increased iron deposition in deeper layers of the motor cortex.
Response 10: Yes, thank you for your point. We acknowledged that.
Comment  11: Perfusion MRI: hypoperfusion in the left parahippocampal gyrus. Further supporting the involvement of perfusion abnormalities in non-motor areas of ALS, another study observed decreased CBF in the thalamus of ALS-FTD patients.
Response 11: We thank the reviewer for this positive feedback.
Comment  12: Results of Methodological quality assessment: Table 2 was compiled according to AMSTAR 2 recommendations. Table 3 is clear and simplifies the task for authors and readers.
Response 12: We thank the reviewer for this positive feedback.
Comment  13: Overall, the results are written using the latest data, which is of great value to all specialists in this field. However, there are a large number of abbreviations that make it difficult to understand the text and the reader has to look in the text for a complete description of these abbreviations.
Response 13: We have minimized the use of abbreviations and ensured that all abbreviations are defined upon their first occurrence. Additionally, a glossary of abbreviations has been included after the conclusion section for easy reference.
Comment  14: The discussion is short and to the point, strictly based on the results obtained, and runs parallel to the results.
Response 14: We thank the reviewer for this positive feedback.
Comment 15: In discussion, in lines 378-380 “These findings align with the well-established clinical presentation of ALS, which is characterized by progressive muscle weakness and atrophy”. Muscle atrophy develops when the lower motor neuron is damaged, and the above changes are signs of upper motor neuron damage.
Response 15: We have revised this statement to clarify the distinction between upper and lower motor neuron pathology.
Comment  16: The “Conclusion” logically follows from the results of the study and is fully consistent with the purpose of the study. However, try to shorten it please.
Response 16: We have shortened the conclusion to make it more concise while retaining the key points. The revised conclusion emphasizes the clinical relevance of the findings and their potential to improve ALS diagnosis and management.

Round 2

Reviewer 2 Report

Comments and Suggestions for Authors

Thank you very much for the opportunity to review this manuscript again.

As I have stated previously, the authors of this review examined the latest systematic reviews in this area and demonstrated the cutting edge of MRI in diagnosing ALS.  The topic is very relevant due to the lack of scientific works devoted to the characterization of morphological changes in the brain of patients with ALS. Undoubtedly, the progress of MRI devices and the development of MRI diagnostic tools and algorithms can reduce the time of early diagnosis and increase the prognostic capabilities of ALS.

The title in the revised version is more consistent with the content.

The introduction is expanded to include a discussion of previous reviews and their limitations. This additional information strengthens this section and makes it more complete.

In my opinion, the purpose of this study in the new version is more consistent with the content and conclusions drawn.

Changes to Figures 1 and 3 have added greater clarity to the Materials and Methods and Results sections.

Conclusions follow logically from the results of the study and are fully consistent with the purpose of the study. I think the changes made to the discussion and conclusions in the new version make the manuscript more complete and scholarly.

Overall, I am very grateful to the authors for their highly professional work in improving the manuscript. I think that the authors have adequately addressed the comments made by the reviewers in the revised version of the manuscript. Therefore, I have no further comments.